# Adaptability Challenges for Organic Broiler Chickens: A Commentary

**DOI:** 10.3390/ani12111354

**Published:** 2022-05-25

**Authors:** Monica Guarino Amato, Cesare Castellini

**Affiliations:** 1Research Centre for Animal Production and Aquaculture, Council for Agricultural Research and Economics, Via Salaria, 31, 00015 Monterotondo, Italy; 2Department of Agricultural, Environmental and Food Science, University of Perugia, Borgo XX Giugno 74, 06124 Perugia, Italy; cesare.castellini@unipg.it

**Keywords:** organic poultry production, adaptability, sustainability, animal welfare, broiler health

## Abstract

**Simple Summary:**

Organic poultry shows an increasing productive trend, rising from 3% in 2017 to 8% in 2019. Regulation EU 848/2018 puts great emphasis on the ability of broilers to adapt to outdoor systems as being essential for organic production. Organic poultry operators meet with regulatory constraints, consumer concerns, and challenges in terms of nutrition, welfare, health, and sustainability. The present commentary considers recent studies on and innovations in these topics that can affect organic production in addition to recent studies on animal adaptability to this production system. It reflects on the concept of broiler adaptability to organic systems not only as a classic genotype–environment interaction but as a necessary prerequisite for facing these relevant challenges.

**Abstract:**

As organic and conventional poultry production increased in the last decade, so did consumers’ concerns, sustainability requirements, and animal welfare as well as health issues. According to Reg. EU 848/2008 on organic production, poultry must be adapted to organic outdoor systems and cope with all the regulatory constraints in terms of nutrition, health, and welfare. Adaptability must take into account the above challenges, constraints, and concerns. Chicken adaptability should not only mean being able to use pasture and outdoor areas, but also mean being able to overcome, or be resilient to, the challenges of organic farming without compromising welfare, performance, and product quality. This commentary identifies solutions to the new challenges that organic poultry chains must face in future productive scenarios, detects consumer viewpoints to provide a perspective on organic poultry production, and summarizes as well as defines chicken adaptability to organic production, assessing the main factors of chicken adaptability.

## 1. Introduction

It is estimated that the rise in the world poultry meat consumption will have a clear positive trend. Consumers are attracted to poultry products due to lower prices, product consistency and adaptability, higher protein and lower fat content, and the lack of religious issues [1].

According to Eurostat, the European Union (EU) produced 13.6 million tons of poultry meat in 2020; 90% of broiler chickens are raised in intensive indoor systems, around 4–5% in less-intensive indoor systems, up to 5% in free-range systems, and 1% in organic systems [2].

European public policy includes organic farming systems in the European “Farm to Fork Strategy for a fair, healthy and environmentally friendly food system” within the wider EU Green Deal [3]. This strategy, alongside a positive market image, determines a positive trend for organic animal production, with a growth rate of 8% in 2019. This growing trend continued throughout 2020 and is expected to increase in 2021.

Organic poultry shows the same increasing trend: EU poultry farms certified as organic rose from 3% in 2017 to 8% in 2019. In addition, organic poultry production is expected to further increase [4] as an alternative to conventional poultry production.

In the EU, alternative broiler production (i.e., free-range and organic) often uses slow-growing genotypes, which are estimated to be around 2 to 5% of the total broiler amount [2]. Slow-growing genotypes are generally preferred for their ability to cope with organic rules while maintaining successful health and welfare states [5].

In organic poultry production (OPP), access to large outdoor areas is mandatory. Regulation EU 848/2018 became operational on the first of January 2022, and puts great emphasis on the requirements for outdoor areas as well as on the ability to adapt to organic conditions, demonstrated by, for example, the ranging and foraging behaviors of chickens.

In addition, the organic poultry sector must move within regulatory constraints, consumer concerns, and low-input agriculture, facing different challenges in terms of nutrition, welfare, health, and sustainability.

This article intends to look at the future challenges of the organic poultry sector in terms of some important topics, such as nutrition, welfare, health, and sustainability, underlining the crucial importance of chicken adaptability to OPP. It intends to review the current knowledge on and definition of adaptability, considering the above challenges, constraints, and concerns.

In this article, the concept of adaptability is considered in a wide sense as the ability of broilers to cope with the whole organic world, consumers included (Figure 1).

In detail, the goals of this commentary are as follows:-Identify solutions to the new challenges that organic poultry chains must face to adapt to future productive scenarios (nutrition, enrichments, etc.);-Detect consumer viewpoints to provide a future perspective to OPP;-Summarize and define chicken adaptability to OPP, assessing the main factors of chicken adaptability.

## 2. Methodology of the Literature Review

Scientific papers were selected using the Web of Science™, Scopus^®^, and Google Scholar databases. Institutional communications of the European Union, the FAO, and the Italian Ministry of Agriculture (*n* = 8), as well as one document from a market analysis company, were also considered.

The search was conducted using the following research strings in articles’ titles, abstracts, and keywords: “organic poultry, poultry adaptability, chicken alternative feeds, health in organic and conventional farm, broiler welfare, chicken behavior in extensive farming, sustainable poultry production, alternative protein feeds”. The search was restricted to the period of 2000 to 2022.

A total of 157 articles were collected, and all of them were analyzed; however, only 95 documents were deemed relevant to the topic of this review and therefore chosen. Most of the 95 studies were on conventional broiler production or chickens as a farming animal. Only 33 documents concerned organic or free-range farming, and 13 studies compared organic and conventional systems. The topics of welfare, resilience, and sustainability increased mainly in the last two decades.

Most of the articles and scientific productions collected came from Europe, followed by the Americas (USA, Brazil, and Mexico) (Figure 2).

Figure 3 shows the distribution of the articles divided by topic. The main topics were directed towards welfare, consumers, and adaptability; for this reason, in some topics (e.g., nutrition) the review was limited to the very latest scientific results connected with adaptability and sustainability.

For other topics, such as the environment, since the review was oriented to organic production, all of the scientific papers that only referred to intensive production were excluded.

### 2.1. Nutritional Challenges

The growth of poultry production has been accompanied by structural changes, characterized by the emergence and growth of “land-independent” productive chains in addition to the intensification and concentration of poultry operations, which determine that conventional farms purchase most of the feed that they use [6].

On the contrary, OPP philosophy should be based on circular agriculture and few synthetic inputs, resulting in animals being more closely linked to a farm.

A study, carried out on a group of consumers, asked for the definition of the “ideal chicken farm” and showed that 27% of the participants mentioned that the ideal chicken farm should use regional feedstuff, preferably grown on the same farm. Although most participants were not aware of what feedstuff is fed to chickens, 23% of the participants highlighted that the feedstuff used should “not be imported”, “not come from South America”, and “not be genetically modified” [4]. Except for the availability of corn and soybean, where the EU is dependent on imports and is far from self-sufficient, organic production is almost entirely faced with these expectations. However, the feed costs of organic diets are about 35–40% higher than those of conventional ones [7].

In organic poultry farming, the feed covers an important role, as the dietary requirements of poultry are very specific and different from those of ruminants [4]. Poultry meat production is strictly tied to feed quality, because broilers grow quickly and have specific requirements for energy and essential amino acids (EAAs). In particular, lysine and methionine [8] must be directly provided by the feed since the birds cannot synthesize them. Methionine is an essential amino acid important in feather growth, protein synthesis and breakdown, and feed efficiency, and impacts egg weight, rate of laying, and immune response [9]. At the same time, lysine is particularly important for chicken muscle development, especially for the breast [10].

Poultry diets are corn and soybean-meal-based, displaying a high energy concentration, low fiber levels [8], and high-quality protein, mainly represented by soybean. EU rules for OPP do not allow the use of ingredients derived from extraction with chemicals (i.e., soybean meal, see next paragraph) or synthetic amino acids. Synthetic solvents are also not allowed in Canada (Organic Production Standard “CAN/CGSB-32.310-2020”), in the USA (“The National List of Allowed and Prohibited Substances” of the USDA Organic Regulations, published by the US Department of Agriculture (https://www.usda.gov/media/blog/2020/10/27/organic-101-allowed-and-prohibited-substances?Page=, accessed on 5 May 2022), or in Australia (National Standard for Organic and Bio–Dynamic Produce, edited by the Australian Government). Lysine and methionine are allowed for monogastric animals in Canada and the USA, but according to the governmental portal of the state of California (https://www.gov.mb.ca/government/index.html, accessed on 5 May 2022) in the USA, some organic certifying organizations do not allow the use of synthetic amino acids. In Australia, only methionine is allowed for poultry.

It is maintained that the need for huge quantities of soybean has caused deforestation; furthermore, the long transport distances of imported soybean meal enlarge CO_2_ emissions. Although organic “deforestation-free soya” is available and used in the EU (about 10 million t in 2018, around 23% of that required), the majority of soybeans come from the Americas. In 2018, the EU imported 15.5 million tons of soybean and 18 million tons of soybean meal from Brazil, followed by the USA and Argentina. In the European continent, soybean is mainly produced in Eastern (mainly in Russia and Ukraine—71% of EU production) and Southern Europe [11]. The current Russia–Ukraine war renders this production and this market much more unstable and uncertain.

Other protein sources could be used in organic diets, and it should be remembered that animal proteins have generally higher biological value than plants.

Accordingly, fish meal could be a protein source for monogastric feeds; however, the exploitation of the seas has led to higher prices, as well as scarcity and sustainability issues. Therefore, to reverse this there is a need for effective, safe, and sustainable protein alternatives. Variation in availability, or limited agricultural land area as well as competition for these protein ingredients, lead to high and volatile prices.

Competition for these resources will increase further with the predicted growth of the human population, since high demand for protein-rich agricultural commodities is expected [12]. Therefore, protein sources that do not compete with human food are highly desirable [13].

Many plants are high in protein: legumes, for example, are rich in protein with a suitable amino acid profile, but also have antinutritional factors (i.e., trypsin inhibitor, lectins, vicine, and convicine) that require heat treatment or other technological processes before being fed to poultry. It is well-known that some crops, such as soybean and canola, are raised for oil, and a concentrated protein meal is produced after the oil has been extracted. However, EU regulations for organic feeds do not permit the use of ingredients where chemical solvents as well as synthetic additives have been used. Other pulses, grains, and seeds (i.e., sesame, sunflower, beans, etc.) provide protein and methionine; even more high-methionine corn hybrids are also being developed [14].

Gluten feed is a key ingredient for the organic diets of chickens. The market of gluten was estimated to be growing at an annual rate of 5.7%, potentially reaching USD 1.24 billion by 2026, showcasing high potential in the mentioned forecast period [15].

Algae are high in protein and methionine, and have the benefit of large amounts of omega-3 fatty acids [16], but are very expensive; additionally, cultivation techniques for them are not yet well-developed.

It must be remembered that, although poultry are able to utilize and digest most amino acids coming from forages, including methionine and lysine [8], grass intake should be mainly considered as a qualitative supplement, and does not represent a relevant source of macronutrients (energy, protein). On this topic, the meat of foraging chickens has less total cholesterol, more vitamin A and E, high *n*-3 levels, and a better n-3-to-n-6 ratio [17].

Although feed costs and restriction rules imply that protein sources mainly derive from vegetables, it is difficult to formulate balanced organic poultry diets using only the available vegetable sources, which do not have high protein levels [9].

To this end, alternative protein sources (i.e., earthworms and insects) could represent interesting feed sources for OPP, but several points (safety, sustainability, and chemical characteristics) should be analyzed and eventually overcome in the future.

The use of earthworms as a protein source in poultry feeding is an opportunity for providing environmental services, as they have well-balanced nutrient contents (e.g., fatty acid profile) and the same or even a better amino acid profile then soybean or fish meal [18].

Insect meal has a protein concentration higher than soybean and, as is the case in soybean, the most abundant EAA is leucine [13], whereas its lysine levels are generally low. Conversely, the variation in amino acids due to the feed substrates, species of insect, and temperature of development is very high. In particular, the feeding substrate is crucial for the nutrient composition of insects; a high content of lysine can only be obtained when specific rearing substrates are furnished. The same is true for fatty acid profiles: the fat composition largely depends on the substrate employed during the development of larvae.

*Hermetia illucens* is the most common insect reared for feed, but the amount of lysine and methionine in larvae do not cover broiler needs [19], requiring the addition of synthetic EAAs, which, as affirmed before, are forbidden under European OPP rules. To obtain a suitable amount of lysine in the prepupae of black soldier flies (BSFs), a well-balanced amount of protein and fiber in the substrate is needed; rearing BSFs on a standard substrate, based on 75% DM of carbohydrates, results in prepupae with a low amount of lysine [20].

Moreover, to compete with conventional protein sources and become relevant as animal feed, as well as fulfil the globally increasing demand for protein, the cost price of insect meal should be significantly reduced [21].

Alternative animal proteins can also be derived from the sea, such as from starfish, mussels, and seaweeds. The EAAs/total amino acids ratio is similar in fish meal and mussel meal, whereas starfish meal has a lower ratio, rendering it a less-suitable source of EAAs (mainly due to the lack of methionine) [13].

Seaweeds supply very low protein levels and, thus, they are not a good alternative feed for protein. In terms of protein level, they are more suitable as a substitute for wheat, but further studies have to be carried out to assess their sustainability [13].

### 2.2. Environmental Challenges

The progressive changing of environmental conditions and the global warming process require adaptation by an animal during the course of its life [22]. Heat stress is one of the environmental factors that decrease the performance and welfare of poultry, as well as the meat quality. Heat stress produces the overexpression of heat shock factors and proteins in chicken tissues, which change the homeostasis of cells and tissues [23]. These changes can affect the physiology of the body and hence the productive performance of chickens. Indeed, commercial chicken strains can reach a high production level, but being very precocious, their body metabolism is very fast and has poor thermoregulation capacities. In contrast, native backyard chickens or less-productive genetic strains are more adapted to “natural” environments, with a robustness that allows them to survive and reproduce constantly [23].

Indeed, fast-growing chicken strains require extremely “controlled” environmental conditions, which play a fundamental role in assuring high productivity in a relatively small space and time [24]. On the contrary, organic production exposes animals to natural light, natural climate conditions, and different temperatures. It is well-known that high temperatures are environmental stress factors that can badly compromise the welfare, health, and production of broilers. In tropical countries, a high temperature represents one of the most significant factors that reduce the performance of broilers [25] and hens.

There are agroecological systems in which integrating crops (herbs and plants) with animals could be suitably linked in OPP to reduce some negative effects of natural environments. The study and design of specific agroecological integrated systems, such as chicken-pastured orchards, will permit the rendering of environmental complexions closer to the behavioral characteristics of chickens [26].

Furthermore, pastured poultry production should be managed to avoid potential impacts on proximal watersheds caused by the runoff of nutrients and microorganisms deposited on pastures, and farmers need to know how their current practices impact the environment [27]. More scientific evidence is needed to effectively support the “environmentally friendly” perception of pastured poultry farming [27]. According to EU directive (91/676/CEE), the chicken density in OPP is calculated to not overwhelm 170 kg N per ha, but a not-homogeneous dispersion of chickens and chicken droppings could concentrate higher amounts of N in some outdoor areas. For this aim, as previously cited, the environmental enrichment of outdoor areas with trees, bushes [28], and the use of active as well as foraging chickens is an important factor in avoiding a high concentration of animals. Generally, non-ranging chicken strains stay close to the house and concentrate droppings in small areas, increasing the nutrient load [29].

In an intensive broiler production system, light is a crucial factor that can positively or negatively affect meat production and welfare. In a natural environment on a sunny day, the intensity of the light in a poultry house can reach 190 lux or more [30], and in free-range settings it can be as high as 100,000 lx. According to manuals for commercial genotypes, the light in meat production should be at 20 lux (https://en.aviagen.com/assets/Tech_Center/Ross_Broiler/Ross-BroilerHandbook2018-EN.pdf, accessed on 5 May 2022), but without artificial light it can sometimes be less, until 5 lux [30].

In OPP, the environmental conditions, such as low density and access to large open spaces, should increase activity levels and improve the leg health of chickens. The provision of natural light led to a reduction in the percentage of birds lying down and also in the number of resting birds [31]. However, the effect of natural light on the activity and behavior of broilers also depends on the genotype; the effect is relatively small on fast-growing broilers, which became very inactive towards the end of the production period [31].

The effect of natural light on the bone development of chickens is also connected to the role of sunlight in making vitamin D. In fact, it is important to emphasize that vitamin D optimizes intestinal calcium and phosphorus absorption for the proper formation of the bone mineral matrix [32]. Conversely, there is little scientific information on the effect of OPP on vitamin D_3_ synthesis and in bone development. Kuhn et al. (2014) found higher vitamin D levels in eggs of free-range hens, and a similar trend could also be expected in meat-type chickens [33].

The interaction between day length and broiler genotype was studied by Schwean-Lardner et al. [34]; they found differences only in terms of the amount of consumed feed in the first month of life.

### 2.3. Health Challenges

Robustness in farm animals was defined by Knap [35] as “the ability to combine a high production potential with resilience to stressors, allowing for unproblematic expression of a high production potential in a wide variety of environmental conditions”.

In this view, in OPP, the robustness and health of animals are considered as being strictly dependent on the capacity of organisms to react to environmental challenges (i.e., temperature and microbial pressure).

Robustness and disease resistance can be improved by selective breeding programs using both genomics and a new quantitative genetic theory to improve the response of birds to environmental changes as well as to predict adaptations. Adaptation could mean better animal welfare, since the animals fit better with the environment and birds suffer lower stress [36].

Salmonellosis remains one of the most frequent foodborne zoonoses, constituting a worldwide major public health concern [37]. Poultry and poultry products are commonly associated with Salmonella, and interventions during production and processing are necessary to manage the risk of infection due to the consumption of poultry products [38].

At the same time, a growing consideration has also developed in regard to antimicrobial resistance (AMR), which is also an emergent problem in animal production. Accordingly, foodborne risks should be faced with an increasing demand for organic and antibiotic-free poultry production.

Pesciaroli et al. (2020) [39], comparing conventional vs. OPP, found a reduction in the burden of antibiotic-resistant commensal *E. coli.* Another study, conducted by Bailey et al. [39], determined the difference in the AMR of *Salmonella* isolated from conventional or organic chickens. The results of this experiment showed that organic chickens were associated with a higher level of *Salmonella* during the early processing steps. However, it was always the case in the Bailey et al. experiment [38] that no difference in the prevalence of *Salmonella* was observed between organic and conventional carcasses post-chill. These observations indicated that, in OPP, interventions during processing can abate contamination risks. In fact, Bailey et al. [38] observed the following: in organic chickens, higher levels of Salmonella were observed during the early processing steps, but no difference in the prevalence of Salmonella was observed between organic and conventional carcasses post-chill.

The results of a study comparing large-scale organic broiler farms vs. conventional broiler farms within the same company in North Carolina suggested that the prevalence of fecal Salmonella was lower in certified-organic birds than in conventionally raised birds, and that the prevalence of antimicrobial-resistant Salmonella was also higher in conventionally raised birds than in certified-organic birds [40].

Overbeke et al. did not find significant differences in the prevalence of Salmonella between organic and conventional broilers at slaughter, but the prevalence of *Campylobacter* was higher in organic flocks [41]. Colles et al. did not confirm that a free-range environment is a major source of infection for free-range broiler chickens [42].

In Sweden, Hansson et al. [43] found that *Campylobacter jejuni* isolated from conventionally raised chickens showed a higher occurrence of resistance to the quinolones nalidixic acid and ciprofloxacin compared with *C. jejuni* isolated from organic chickens. This difference between isolates from different production systems agrees with findings in other studies on organic and conventional poultry as well as poultry meat [43]. Williams et al. [44] did not find any detectable difference in *Campylobacter* carriage in fast- and slow-growing birds, but artificial infection with *C. jejuni* affected the incidence of hock marks and pododermatitis, which was greater in the fast-growing breed than in their slower-growing counterparts.

Another study compared samples of organic broilers and conventional broilers collected from a processing plant in the Midwestern US, and found no significant differences in terms of AMR prevalence between organic and conventional isolates [45].

Regarding avian flu (AF), it should be emphasized that, on the one hand, it is maintained that OPP is more vulnerable to AF due to the possible contact with wild birds; however, on the other hand, it seems that the intensification of the poultry sector [46] increases the emergence of AF.

There is not detailed information on the relevance of coccidiosis in conventional and OPP farms; however, it seems that, in OPP, coccidiosis protection increasingly relies on vaccines [47] being an important alternative to drugs.

The studies cited above and their results may lead to the conclusion that the impossibility of achieving high biosecurity measures in free-range systems may not be a severe threat to human health; however, disease prevention is important and critical in organic poultry production.

In addition, the large outbreaks of avian flu in the US (https://www.cdc.gov/flu/avianflu/index.htm, accessed on 5 May 2022) and in Europe (https://www.efsa.europa.eu/en/topics/topic/avian-influenza, accessed on 5 May 2022) result in stricter controls on biosecurity measures, as is also the case in organic livestock farming.

### 2.4. Sustainability Issues

Animal feeding is the main factor responsible for environmental impacts, independent of the rearing system (cage and non-cage; conventional and organic) [48], mainly in monogastric animals where digestion produces minor emissions of greenhouse gases.

Therefore, the productive performance (daily weight gain or egg production/day), the feed intake, and, consequently, the feed conversion ratio (FCR) have been identified as highly influencing the environmental impact of the poultry supply chain. Therefore, the search for the continuous on-farm improvement of these traits is always mentioned as the first aspect to pursue in order to obtain more sustainable production. Accordingly, interventions linked to the protein requirements and/or on the protein source itself have a great influence on environmental impacts [48].

In summary, the environmental sustainability of the poultry diets depends on four main factors:
The FCR of the chicken (how much feed is necessary for producing one unit of food);The feed ingredients used (i.e., different crops need more or less inputs);The cultivation techniques of the crops (i.e., in OPP chemical fertilizers are not allowed, which is also the case for herbicides and other chemical drugs);The dietary requirements (energy, protein, amino acid profile, etc.) of the genetic strain.

#### 2.4.1. FCR

Considering the good feed conversion efficiency of high-performance broilers, some authors [49,50,51] found that intensive systems are more sustainable than organic or free-range ones. On the other hand, Castellini et al. and Boggia et al. showed that free-range chickens have a lower environmental impact [52,53]. These discrepancies mainly depend on the genetic strain and the feed ingredients adopted in the different systems. Accordingly, the fact that the free-range and intensive systems have relatively close scores reflects the current opinions of scientists about these systems.

At the same time, in OPP it is possible to combine crop and livestock production (see Section 2.3), and the best performance of the integrated system is consistent with the emerging literature. In the study of Rocchi et al., the best performance of the integrated system is explained by the environmental and economic benefits achieved by combining free-range animals with trees or orchards [54].

#### 2.4.2. Feed Ingredients

According to the Farm to Fork Strategy [55], food waste generated in the EU represents “about 6% of total EU emissions”. This means that we should accelerate the efforts to reduce these emissions, including by safely upcycling certain side streams. Based on these studies and reports, the sustainability of poultry production is strictly correlated with the nature of feeds, and although genetic selection has the potential to reduce the resources needed for body growth, the need for certain feed ingredients (mainly protein sources) to fulfil broiler requirements may limit the path to sustainability. The use of alternative feed ingredients, such as locally grown protein crops, alternative sources (i.e., insect meal), and byproducts, as a replacement for soybean can potentially reduce environmental impacts, deforestation, and CO_2_ emissions [51]. A relevant aspect of the resources related to feeds and food is the risk of poultry competing with humans for protein sources. The 100% organic feeding should be based on an increased amount of locally sourced feedstuffs, which must not compete with human food or require transport. This can lead to increased difficulty to meet birds’ required EAAs (see also Section 2.2) without overfeeding and consequently increasing the excretion of N in the environment [56].

Land use is one of the most demanding impact categories for OPP [53]. It should be mentioned that, in the EU, and mainly in Italy, the marginal lands have increased [57]. Therefore, a wider use of marginal land (where it is expected to create OPP) and conservative agricultural techniques could be considered a plus for the maintenance of territory integrity, ecosystem services, and biodiversity.

In this context, the sustainability of fish meal in poultry diets can be a concern due to fishing practices and their potential impact on wild fish populations [9]. Fishing often takes place in areas far from where poultry are raised, and is not based on the cycling of local nutrients. In addition, overfishing has resulted in limited fish meal availability, and stewardship to protect fish populations is critical [9].

As previously affirmed, insects and other invertebrates could replace fish meal in organic diets. Insects, due to their high reproductive potential, nutritional quality, low water and space requirements, ability to use waste from vegetable production as feed, and low environmental impact, can theoretically produce feed for livestock in a more sustainable way [9].

Within a sustainable context, insects should use un-utilized large biomass, such as waste or byproducts. However, current legislation limits the use of raw materials that can be used as insect substrates. For example, in EU legislation on feed hygiene and traceability insects are considered livestock, and the use of catering waste is forbidden [13]. Moreover, the growth of larvae requires a high environmental temperature; thus, insects should be reared close to cheap heat sources such as diesel generators, biogas power plants, etc.

#### 2.4.3. Cultivation Techniques

In organic agriculture, all of the inputs employed in agricultural production (fertilizers and products for controlling weeds, insects, bacteria, and viruses) should be natural. Accordingly, only organic manure and “natural” pesticides (i.e., minerals such as Cu, S, etc.) are permitted. This fact renders the protection of crops more difficult, but generally, in terms of the amount and quality of energy inputs, OPP requires less and more-renewable energy (i.e., every t of urea requires about 1 t of oil), because fewer fossil fuels are used.

#### 2.4.4. Requirements

As stated already, it is hard to formulate balanced organic diets for broilers, as soybean meal and synthetic EAAs are forbidden. To this end, the use of slow-growing strains also means lower dietary requirements, which are more easily complied with by using an organic formulation. Accordingly, the use of less-concentrated diets with a higher presence of “local” ingredients is often able to cover the lower nutritional requirements of these strains [58].

In addition to the feed sources employed, the amount of energy used in the conditioning and management of animals also plays a fundamental role because poultry houses need a great amount of energy. Different weather conditions, different features of the animal farming (e.g., stocking density), and different building characteristics may modify these values [59]. Comparing alternative and conventional rearing systems of chicken, one study showed that the free-range system had the same or lower environmental impacts when compared to the conventional one, at least when the feed conversion ratio of the birds is not significantly increased and when modern energy-efficient housing systems are applied [60].

### 2.5. Welfare and Consumer Perceptions

Animal welfare is at the forefront of consumers’ minds and perceptions. The recent change in consumer feeling for meat products has led to the development of more welfare-friendly poultry production systems [61]. From an animal welfare perspective, the “social” and institutional sustainability aspects of broiler production require great attention [62].

According to the Agriculture and Horticulture Development Board (AHDB) (https://ahdb.org.uk/, accessed on 5 May 2022), when choosing meat products, animal welfare is claimed to be one of the most important factors for consumers after other considerations such as presentation and price, and it comes before environmental considerations.

It is well-known that in many countries consumers are giving more importance to ethical issues, such as animal welfare, animals’ quality of life, and positive experiences for animals. Consumers usually associate housing conditions with animal welfare [4,63], and when thinking about positive animal welfare they think of small farms with animals living outdoors in a natural environment [63]. The “Ideal Chicken Farm”, as described in the focus group discussion described by Escobedo del Bosque et al. [4], is a free-range farm with chickens walking with ample space on green pasture. In this view, organic poultry farming strongly meets these expectations, because OPP uses open areas and lower chicken density.

It should be emphasized that the estimated improvement in animal welfare in OPP is achieved only if the right genetic chicken line is chosen [5,54,64,65,66]. As already shown in Section 2.1, fast-growing broilers at an older age (81 d) are very heavy and have an unbalanced body conformation as a result of intense genetic selection for additional breast muscle and body mass, which render kinetic activity more difficult and unusual. Many studies have recommended the use of slow-growing strains in OPP, which do not have the same welfare problems of the current commercial strains [5,65,67]. Slow-growing strains have lower mortality, less incidence of leg weakness and cardiovascular diseases, and generally show an improved welfare status [68]. In any case, not all slow-growing strains, even those with similar weight gain, have the same adaptability to OPP (see Section 2.6).

Despite animal welfare being a main concern for consumers, the same consumers often unconsciously categorize chickens as a commodity, devoid of authenticity as a real animal with an evolutionary history and phylogenetic context [69]. Indeed, the idea of chicken psychology and intelligence is a strange concept for most people. Chicken intelligence appears to have been underestimated and overshadowed by other species of birds. This asymmetry in the literature is likely the reflection of the disconnection between current scientific knowledge and public opinion [70]. According to Marino [69], chickens possess a number of visual and spatial capacities, communication skills, and a capacity to reason and make logical inferences; they perceive time and have a complex of emotions, empathy, and a distinct personality, as is the case for other birds and mammals.

Therefore, these facts should be considered when talking about chicken welfare. Yeates, in a study in 2017 [70], suggested combining consumers’ opinions, expressed by an association of citizens that care about animal welfare (e.g., the Farm Animal Welfare Council’s concept of a good life), and scientifically-based welfare measures (e.g., the EU Welfare Quality project). The use by farmers and technicians of welfare self-assessment tools based on these principles and codesigned by researchers, practitioners, and citizens could also help to improve on-farm welfare. Their recognition by official welfare labeling standards should also be in favor of the perception of organic production as high quality by citizens and consumers.

According to the above studies and considerations, it may be stated that animals should be reared in the right environment to allow for the expression of natural behavior and to live a life worth living (and, ideally, a good quality of life) [56]. As with most farm animals, broiler production chains are complex, and many factors affect the welfare outcomes. It is vital that outcomes are monitored on an ongoing basis across the whole production cycle. This should not only include the monitoring of negative welfare outcomes, such as lameness, but also indicators of positive experiences, such as engagement with environmental enrichment [62], which is fundamental in organic production.

Finally, chicken welfare during transport and slaughter is also important: different transport practices and lengths can reduce bird welfare and increase the risk of bodily injuries (broken wings/legs and overall distress) as well as mortality [71]. However, few data are available on the effect of transport on birds from free-range systems. It is expected that the more active animals used in OPP will respond to transport stress differently than fast-growing chickens do. In particular, wing shaking determines a reduction in muscle glycogen and an alternate acidification of meat. To reduce the stress of transport, different mobile slaughtering houses have been proposed, which are maintained to only be suitable for small chicken farms [72,73,74,75].

### 2.6. Broiler Adaptation to OPP

Given the challenges described above, it is evident that, apart ontogenetic factors, an organically raised chicken has to adapt to several “external” constraints, some of which are legislative rules, such as the prohibition of synthetic amino acids and allopathic drugs, others of which come from new consumer views and the need to consider animal welfare as a *conditio sine qua non*, and still others that derive from balances linked to world economies, raw material flows, and the need to make production more sustainable.

Naturally, there are also animal-based challenges linked to welfare, health, growth traits, and meat characteristics that can influence the adaptability of chickens to organic production.

In general, animal adaptability concerns the use of available resources (e.g., land, feed, water, housing, and capital) [76]. The adaptability of a genetic strain in various rearing conditions (e.g., intensive, free-range, and low-input) and its stability might be assumed as good indicators for ecological, ethological, and ethical norms in animal production. A breed is said to be stable if, in a given environmental condition, its production and reproduction performances remain more or less constant over different years. Cassandro et al., in 2014, studied four local breeds reared in an Italian region in different environmental conditions. Despite the fact that the four breeds are usually reared in the same region and under different environmental conditions, the results showed different adaptabilities to environments and stability indexes for fertility traits [77]. Broadly, adaptation is a short-term and long-term response to a challenge (stressor) intended as specific environmental condition. A short-term response, although linked to a phenotypic reaction, has a clear genetic basis, which affects the different abilities of animals to cope with environments when exposed to the same stressors [76]. For example, some management strategies cause a response that is strictly influenced by genetics, e.g., high-production cows react to heat stress compared to adapted local breeds even if some mitigation strategies are adopted.

Therefore, the adaptation of animals to specific environments can also be improved by genetic selection. Unfortunately, for economic reasons, selection was often concentrated on a few specific traits, irrespective of their relationships to other traits such as behavior, welfare, and meat quality. Thus, the selection for high performance has led to very specialized lines which are undesirable in some different scenarios, e.g., OPP [78]. In fact, the genotype–environment interaction is often more relevant than the effect of the genotype itself, and thus, since the majority of the genotypes are selected for an intensive system—e.g., very-controlled environment, high-quality feed, vaccines, etc.—this renders scarce the number of strains available for OPP. The best genotype in one environment may not be the best in another environment due to the genetic sensitivity to macro-/microenvironmental stimuli [67,79].

It should be emphasized that highly productive strains generally have low available resources for adaptation (allocation resource theory [80]). FG strains are selected for their high productive performance and to direct nutrition resources towards muscle growth or egg deposition. At the same time, active behaviors, immune responses, and thermotolerance were reduced [23]. The selection for high production performance modified behavioral strategies: activities with high energy costs (i.e., foraging and social interactions) show lower frequency in highly productive animals, allowing them to save energy that can be reallocated to production traits. Accordingly, in OPP, where there are required “natural” behavioral traits (kinetic, thermotolerance, and immune responses), these high-performance strains are non-adapted in principle.

In contrast, productive performance is also often given primacy in the case of adaptability studies. In Nigeria, the adaptability of different chicken strains to a high temperature only took into consideration production parameters, such as feed intake, body weight gain, and feed conversion ratio [81]. A breed is considered adaptable to an environment if the performance (averaged over years) varies little across environments [77].

Animal–environment connection and interaction are central in the concept of resilience, which is the capacity of animals to cope with short-term perturbations in their environment and return rapidly to their pre-challenge status [22]. Resilient animals are animals that need little/less attention; increasing resilience is therefore desired [82].

Adaptability and resilience are connected with robustness: the ability of animals that combine high production potential with resilience to external stressors, allowing the expression of high production potential in a wide variety of environmental conditions [35] without losing their integral structures and functions [83].

A study on the assessment of an adaptability score of chickens to organic farming, which included behavior, plumage condition, and body score, found that genotypes had a strong effect on the adaptation score, and that this score was affected by the interaction of different factors, such as behavior, productive performance, and welfare [67]. It should be noted that, although a slow growing rate is a prerequisite of adaptation, the genotype seems to be a specific and independent factor of adaptability, whereas the daily weight gain has less predictive power [84]. Accordingly, the daily weight gain should be considered a prerequisite, but the ability to adapt to environmental and structural conditions should be assessed by characterizing the single genotype.

In terms of adaptability, an important trait is surely the foraging behavior. Foraging involves pasture and kinetic attitudes. Ferreira et al. [85] found that that the foraging of chickens was correlated to range use from an early age and during all of the rearing periods, independent of the season. According to their study, foraging was the only behavior that showed within-individual consistency, and therefore may serve as a useful behavioral predictor of range use in free-range broiler chickens. Only active chicken lines are able to use the outdoor areas, whereas sedentary lines prefer to stay inside a barn and spend almost their whole lives eating and resting, with severe welfare issues.

As cited before, as part of the average adaptability of a specific chicken strain, chickens possess a number of capacities and have emotions, distinct feelings, and personalities, as is the case for other birds and mammals. The differences in behavior on the free-range in organic systems may depend on the way each chicken recognizes and processes its environment information in relation to its personality and cognition capacities. Indeed, Ferreira et al. [86] have shown that chickens have stable ranging behavior over time, and that the chickens defined as high rangers had less capacity to improve their spatial memory than low rangers when submitted to successive spatial tests. Furthermore, high rangers of a slow-growing chicken strain forage more than the low rangers, with individual consistency over time [85]. The low rangers were also more prone to be near their near conspecifics than high rangers during social motivation tests [87]. This ranging behavior could also be linked to a bird’s emotional state, since in laying hens a longer tonic immobility (TI) duration during TI tests, used to evaluate fearfulness, was associated with less time spent outside by a bird [88].

In addition, bone health and gait scores can also be affected by the genotype. Slow-growing and active genotypes have better gait scores, better tibia conformation, and less tibial dyschondroplasia [65,67,89,90,91]. A study that compared the breast and gastrocnemius fiber muscle of two fast-growing lines reared in different housing conditions found that pathological changes were mainly seen in birds with outdoor access, and were indicative of their worse adaptability to the outdoors [92,93].

Several authors investigated the genetic ability of chickens to utilize open air for movement and pasturing [17,84,88,90,91,94,95,96], underlining the differences that the activity of genotypes brought to different welfare and meat traits. Meat from pasture-raised chickens may present nutritional benefits, such as flavor, and contribute to positive sensory attributes of meat [97]. On the contrary, access to pasture did not affect meat quality parameters when fast-growing chickens were reared [98].

Therefore, slow-growing broilers are more suitable for free-range systems [99], even if the physical properties of meat do not seem influenced by the access to open runs. On the contrary, differences in terms of poly-unsaturated fatty acid (PUFA) were identified in active genotypes: they have a higher amount of n-3 and n-6 PUFA and a lower amount of α linolenic acid, utilized to produce longer-chain fatty acids such as eicosapentaenoic (EPA 20:5 n3) and docosahexaenoic (DHA 22:6 n3) [97,100]. These differences are greater in spring, when green pasture is abundant. Recently, Cartoni et al. [101] found that a sedentary (fast-growing) genotype uses a great proportion of its energy intake to increase its body growth at the expense of other bodily functions.

A study compared a highly selected meat-type broiler reared with and without outdoor access: the results showed that the number of necrotic fibers was higher in chickens with outdoor access, which confirms that it is more difficult for this genotype to adapt to outdoor environmental conditions [92].

Pasture and free-range systems potentially improve gut health, indirectly also promoting animal welfare. Indeed, free-range production not only provides space, fresh air, and direct sunlight, but allows birds to express natural behaviors, such as dust bathing, scratching, foraging, running, and flying, while reducing incidences of pecking due to low stocking density [102].

In terms of time spent outdoors, it could be argued that free-range and organic systems are no different to the conventional one if the chickens (i.e., sedentary chickens) never leave the housing [8] and rest for most of the budgeted time [31,84,90,103].

Moreover, the discrepancy between the current living conditions of grandparental and parental lines reared indoors and the commercial crossbreeds living outdoors should be mentioned. Very soon in Europe this mismatch will be reduced when breeders also have outdoor access, whereas for the moment they are indoors to protect them from health issues; this is a great challenge for practitioners and breeding firms.

## 3. Conclusions

Based on previous studies and the results shown above, adaptability to OPP can mainly be summarized as the ability of a chicken to take advantage of outdoor areas.

However, as we saw in the first part of this article, in a wider sense, organic systems face a series of adaptability challenges. Indeed, the adaptability of a chicken does not only mean being able to use pasture and outdoor areas, it also means being able to overcome, or be resilient to, the challenges of organic farming without compromising welfare, performance, and product quality. Nonetheless, a chicken suitable for organic farming must be genetically adapted to a different feeding plan and must have sufficient robustness to avoid or reduce the risk of diseases that could compromise its welfare, health, and productivity. In addition, organic poultry production is also called upon to contribute to sustainability, meaning that it is necessary to find genotypes that, at the same time, are more efficient, consume less, and have good foraging behavior.

In addition, and in contrast to the natural behavior of kinetic lines that express themselves in outdoor areas, chickens should be adapted to both indoor and outdoor environments due to the recurrent spreading of avian flu.

## Figures and Tables

**Figure 1 animals-12-01354-f001:**
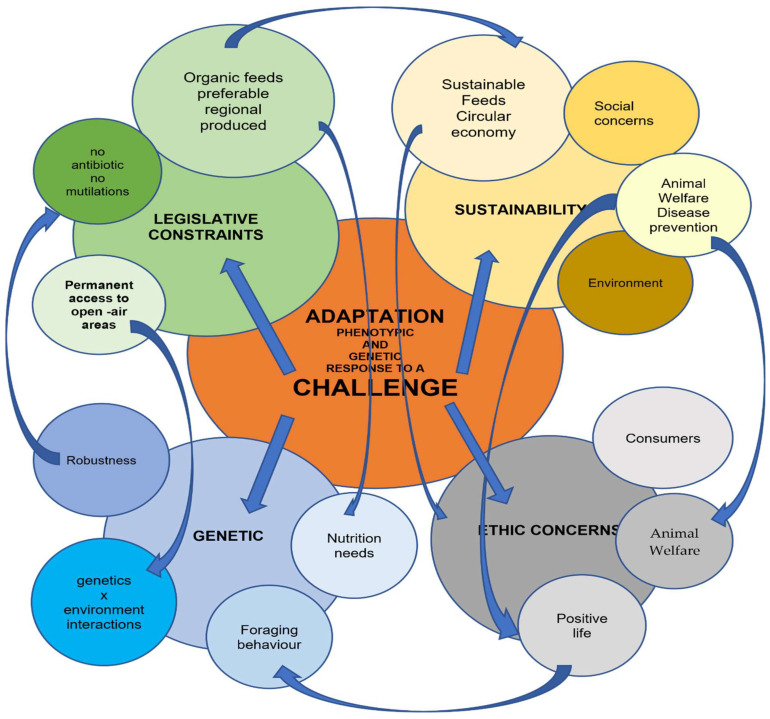
Conceptual map of the concept of adaptability.

**Figure 2 animals-12-01354-f002:**
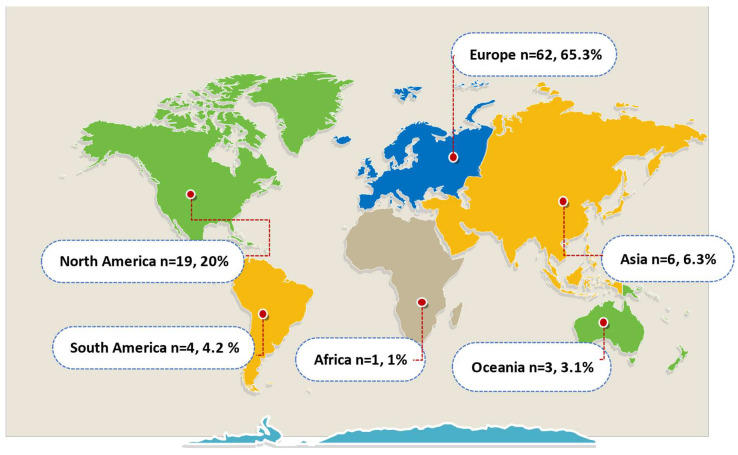
Geographical origin of the article (*n* and % of total articles). Adapted form Simple Word Map https://www.mapchart.net/world.html, accessed on 5 May 2022.

**Figure 3 animals-12-01354-f003:**
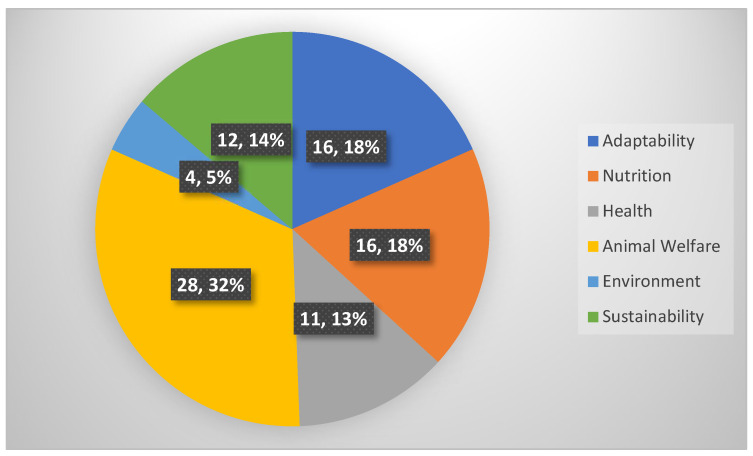
Distribution of revised studies per topic (*n* and %).

## Data Availability

Not applicable.

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
