# Peer review of "Adaptability Challenges for Organic Broiler Chickens: A Commentary"

_animals, 2022, doi:10.3390/ani12111354_

Round 1

Reviewer 1 Report

Accept Minor Revision -

Line 19 - Add a few more sentences of detail to the abstract.

Line 80 - Add two or three sentences to explain poultry adaptability.

Line 132 - The audience for your paper is worldwide.  Recommend adding some information on organic standards in other countries such as the U.S., Canada, and Australia.

Line 199 to 204 - The use of seaweed would probably not be sustainable. Seaweed is a limited resource.

Line 254 - Add additional information and references on the effect of different types of lights on different broiler genotypes.

Line 263 - The references you have for robustness is for pigs.  It would be good to add information for more than one species.

Line 326 - There have been large outbreaks of avian flu in the U.S. These outbreaks have penetrated strict biosecurity protocols.  Add references.

Line 353 - Provide a more specific description on how genetic strain is affected by feed ingredients.

Line 389 - What type of waste is being used as feed. Please explain.

Line 446 - Explain how the birds differ in adaptability.

Lines 494-502 - Very vague overgeneralized description. Provide two or there examples to explain the concepts.

This paper contains information that should be published.

Author Response

Dear Reviewer,

thank you very much for your valuable comments that are going to really improve our study. Hereby you can find our answers and in the revised text your suggestions. According you suggestions about the language, we decide to send the reviewed manuscript to an English Editing Service.

Accept Minor Revision -

Line 19 - Add a few more sentences of detail to the abstract.

Done

Line 80 - Add two or three sentences to explain poultry adaptability.

It was just the word used for the search

Line 132 - The audience for your paper is worldwide.  Recommend adding some information on organic standards in other countries such as the U.S., Canada, and Australia.

Done

Line 199 to 204 - The use of seaweed would probably not be sustainable. Seaweed is a limited resource.

We totally agree, we added a sentence

Line 254 - Add additional information and references on the effect of different types of lights on different broiler genotypes.

Done

Line 263 - The references you have for robustness is for pigs.  It would be good to add information for more than one species.

The reference number 35 is about poultry robustness

Line 326 - There have been large outbreaks of avian flu in the U.S. These outbreaks have penetrated strict biosecurity protocols.  Add references.

Done

Line 353 - Provide a more specific description on how genetic strain is affected by feed ingredients.

Non ho capito cosa vuole

Line 389 - What type of waste is being used as feed. Please explain.

Done

Line 446 - Explain how the birds differ in adaptability.

It is fully explained in §2.6 we put the right reference

Lines 494-502 - Very vague overgeneralized description. Provide two or there examples to explain the concepts.

Done

Reviewer 2 Report

Lines 88-89 “From a temporal point of view, the topics welfare, resilience and sustainability increased mainly in the last two decades”  Change sentence, considering that the period considered is practically two decades.

Figure 2 - 92 documents are reported and not 90 documents.

References

Yeates, J.W. Animals. J. Agric. Environ. Ethics 2017, 30, 23–35, doi:10.1007/s10806-017-9650-2.  title is missing.

 “Okusanya, P.O.; Akinlade, O.O. ADAPTABILITY AND GROWTH PERFORMANCE OF DIFFERENT STRAINS OF BROILER CHICKEN TO HIGH TEMPERATURE VARIATIONS IN NORTH CENTRAL NIGERIA. 2019, 324–328.” Magazine  missing.

Author Response

Dear Reviewer,

Thanks for your valuable comments. We modified the text as suggested.

Comments and Suggestions for Authors

Lines 88-89 “From a temporal point of view, the topics welfare, resilience and sustainability increased mainly in the last two decades”  Change sentence, considering that the period considered is practically two decades.

Done

Figure 2 - 92 documents are reported and not 90 documents.

Corrected the text

References

Yeates, J.W. Animals. J. Agric. Environ. Ethics 2017, 30, 23–35, doi:10.1007/s10806-017-9650-2.  title is missing.

 “Okusanya, P.O.; Akinlade, O.O. ADAPTABILITY AND GROWTH PERFORMANCE OF DIFFERENT STRAINS OF BROILER CHICKEN TO HIGH TEMPERATURE VARIATIONS IN NORTH CENTRAL NIGERIA. 2019, 324–328.” Magazine  missing.

Done

Reviewer 3 Report

The review/commentary here proposed is very well presented and certainly provides a clear perspective about the future directions of poultry farming, especially considering the global trend towards organic breeding. The authors thoroughly considered the mainly important points concerning the imminent evolution of poultry farming. Considering the innovative perspective in approaching such a problem, I would definitely recommend this paper for publication after the revision of the following points.

Minor points

Even if the manuscript is well presented, in many parts the language is wordy, and the sentences/period structures are too complex. I would recommend a thorough additional revision (possibly by a native English speaker) of the entire manuscript to improve the language were needed.

Here few examples of possible amendments:

Lines 31-32 Please rephrase.

Line 63 I would suggest adding a coma after the word “view” and to remove “in wide sense”.

Line 65 “In detail, the goals of this coommentary is to:” maybe better to rephrase as following: “More in detail, the aims of this commentary are to:”. The authors could as well consider to move the preposition “to” to the head of each following bullet point.

Line 112 “A study, carried out on a group of consumers, asked the definition of “ideal chicken…”, maybe better like this: A study, carried out on a group of consumers and asking for the definition of “ideal chicken…

Line 485 “Given the challenges described above, it is evident that a part ontogenic factors, an…” This sentence is not clear, please rephrase.

Line 486-490 All this period is very complex and needs substantial rephrasing e.g. “… some of which are legislative roles…”, I would modify it into “… some of which depend on legislative roles…” or “… some of which are tied to legislative roles…”.

Lines 607-621 I would consider using this part as the conclusion section.

Author Response

Dear Reviewer,

Thank you very much for your consideration and for the positive and valuable  comments. According your suggestion about the language,  we decided to submit the work to an English editing service. 

The review/commentary here proposed is very well presented and certainly provides a clear perspective about the future directions of poultry farming, especially considering the global trend towards organic breeding. The authors thoroughly considered the mainly important points concerning the imminent evolution of poultry farming. Considering the innovative perspective in approaching such a problem, I would definitely recommend this paper for publication after the revision of the following points.

Minor points

Even if the manuscript is well presented, in many parts the language is wordy, and the sentences/period structures are too complex. I would recommend a thorough additional revision (possibly by a native English speaker) of the entire manuscript to improve the language were needed.

Here few examples of possible amendments:

Lines 31-32 Please rephrase.

Line 63 I would suggest adding a coma after the word “view” and to remove “in wide sense”.

Line 65 “In detail, the goals of this coommentary is to:” maybe better to rephrase as following: “More in detail, the aims of this commentary are to:”. The authors could as well consider to move the preposition “to” to the head of each following bullet point.

Line 112 “A study, carried out on a group of consumers, asked the definition of “ideal chicken…”, maybe better like this: A study, carried out on a group of consumers and asking for the definition of “ideal chicken…

Line 485 “Given the challenges described above, it is evident that a part ontogenic factors, an…” This sentence is not clear, please rephrase.

Line 486-490 All this period is very complex and needs substantial rephrasing e.g. “… some of which are legislative roles…”, I would modify it into “… some of which depend on legislative roles…” or “… some of which are tied to legislative roles…”.

Lines 607-621 I would consider using this part as the conclusion section.

Done